# Epidemiological, Physiological and Diagnostic Comparison of *Plasmodium ovale curtisi* and *Plasmodium ovale wallikeri*

**DOI:** 10.3390/diagnostics11101900

**Published:** 2021-10-14

**Authors:** Joseph Hawadak, Rodrigue Roman Dongang Nana, Vineeta Singh

**Affiliations:** 1ICMR-National Institute of Malaria Research, Dwarka, Sector 8, New Delhi 110077, India; johawadak@yahoo.fr (J.H.); na_rodrigue@yahoo.fr (R.R.D.N.); 2Institute of Medical Research and Medicinal Plants Studies, Yaoundé P.O. Box 13033, Cameroon

**Keywords:** malaria, *Plasmodium ovale wallikeri*, *Plasmodium ovale curtisi*, diagnosis

## Abstract

Nowadays, *Plasmodium ovale* is divided into two non-recombinant sympatric species: *Plasmodium ovale wallikeri* and *Plasmodium ovale curtisi*. In this mini review, we summarize the available knowledge on the clinical/biological aspects of *P. ovale* spp. malaria and current techniques for the diagnosis/characterisation of *P. ovale curtisi* and *P. ovale wallikeri*. *P. ovale wallikeri* infections are characterized by a deeper thrombocytopenia and shorter latency compared to *P. ovale curtisi* infections, indicating that *P. ovale wallikeri* is more pathogenic than *P. ovale curtisi*. Rapid diagnosis for effective management is difficult for *P. ovale* spp., since specific rapid diagnostic tests are not available and microscopic diagnosis, which is recognized as the gold standard, requires expert microscopists to differentiate *P. ovale* spp. from other *Plasmodium* species. Neglect in addressing these issues in the prevalence of *P. ovale* spp. represents the existing gap in the fight against malaria.

## 1. Introduction

Malaria is a tropical and subtropical infectious disease caused by protozoa of the genus *Plasmodium* and responsible for 409,000 deaths worldwide in the year 2019 [1]. Among human *Plasmodium* spp., *P. falciparum* and *P. vivax* are the most prevalent species in Africa and the rest of the world, respectively [1]. These two species are involved in the majority of cases of disease severity and therefore the most studied, whereas two other non-falciparum species, viz. *P. malariae* and *P. ovale* spp., are relatively less pathogenic, coupled with a low prevalence rate [2,3]. A steady rise in infection due to *P. ovale* spp. has been observed in several regions of sub-Saharan Africa (sSA) [4,5,6]. Recently, a systematic review highlighted the involvement of *P. ovale* spp. in severe malaria, which is the same as that of *P. vivax* and thus deserves as much attention as the other species [2].

*P. ovale* spp., having two sympatric species, *Plasmodium ovale curtisi* (*Poc*) and *Plasmodium ovale wallikeri* (*Pow*), is the last plasmodial species described in 1922 by Stephens, who named it for the oval shape of some infected erythrocytes [7] commonly found in Africa, Southeast Asia and Papua New Guinea [8,9]. Many epidemiological studies conducted till date do not specifically target either of these two species, though a few have reported on the emergence of *Poc* and *Pow* and the evolution of their involvement in severe malaria [2,10,11]. In addition, the available specific screening protocols for these two species are almost non-existent in health centre laboratories for routine malaria diagnosis. In the year 2014, Fuehrer and Noedl published a review presenting the techniques used for the phylogenetic classification of *Plasmodium* species based on molecular markers and diagnostic techniques of *P. ovale* spp. [12]. Three years later, Zaw and Lin presented a more extensive review adding geographical distribution, molecular discrimination techniques and clinical differences between the *Poc* and *Pow* species [13]. In the previous two reviews, no information on the performance of the rapid diagnostic test (RDT) relevant for *P. ovale* spp. was mentioned. The present mini review focusses on two important aspects: (i) a brief comparison of *Poc* and *Pow* on clinical and biological aspects; (ii) an overview of the currently available techniques for diagnosis and molecular markers for the genetic characterisation of *Poc* and *Pow*, summarizing the available knowledge with respect to these two species.

## 2. Ecological and Biological Mechanisms behind the Distinction of *Poc* and *Pow*

Since the description of *P. ovale* spp. by Stephens in 1922, the two species were considered a single species, until 2010, when Sutherland described *Poc* and *Pow* as two sympatric species endemic in Africa [7,14]. *Poc* and *Pow* are now recognized as two non-recombinant species that diverged about two million years ago and that remain distinct due to a strong genetic barrier [15,16,17]. Three principal hypotheses have been formulated by Sutherland to explain the *Poc* and *Pow* segregation [14]. Although an ecological and/or seasonal differential distribution serving as a physical boundary between the two species was first suggested, current knowledge shows that these two species are sympatric and share the same geographical locations [13]. A second hypothesis stipulates host specificity, which is being contradicted by studies that have reported *Poc* and *Pow* co-infection [18,19]. The most plausible hypothesis till date is of the genetic barrier, i.e., there is an incompatibility between *Poc* and *Pow* gametocytes or the sporozoites obtained from cross-fertilisation between them as they are devoid of the ability to infect [14]. This last hypothesis needs to be thoroughly investigated in all aspects to understand the distinction between the two species despite the limitations as no established in vitro culture technique exists for *P. ovale* spp.

## 3. Epidemiology and Geographic Distribution of *Poc* and *Pow*

*P. ovale* spp. infection accounts for less than 1% of all malaria cases worldwide [1]. Current data on *Poc* and *Pow* species distribution and prevalence are not precise, in the sense that the use of species-specific primers is not yet systematically adopted in all epidemiological studies of malaria. However, of late, more studies are being conducted on the detection and differential characterisation of these species [20,21]. *Poc* and *Pow* have been reported mainly in sSA and Asia [18,19,22,23,24,25,26,27] and their prevalence ranges between 0.1% and 4% in Africa and is less than 0.1% in Asia with no significant difference between *Poc* and *Pow* [19,25,26,27,28]. Moreover, misidentification of an increase in imported malaria cases in non-endemic areas have been reported from Singapore, United Kingdom, Spain, Italy, France and China among travellers, mainly from Ivory Coast, Cameroon, Nigeria, Central African Republic, the Democratic Republic of the Congo and Guinea [20,29,30,31,32,33,34]. *Poc* and *Pow* prevalence is supposedly much higher than that currently reported, as *P. ovale* spp. prevalence (7% to 12%) has been reported without species specification in studies conducted in Zambia, Gabon and Cambodia [5,19,26,35]. This emphasizes the need to popularise the use of more efficient diagnostic and characterisation tools to draw up a real distribution map of these species and to understand their epidemiological dynamics and real contribution to the malaria burden.

## 4. Diagnostic Assays for *Poc* and *Pow*

### 4.1. Rapid Diagnostic Tests

As with any other disease, rapid diagnosis followed by efficient treatment is one of the cornerstones of malaria control and elimination strategies. RDT remains the most affordable, rapid and available diagnostic technique, requiring little field expertise. During the years 2008 to 2019, the efficacy of 332 RDTs from more than 50 manufacturers was evaluated by the World Health Organization (WHO) to ensure the quality of the kits provided [36]. By 2014, the performance of these RDTs was estimated to be around 93% [37] and, to date, almost all of them have conformed to the WHO’s criteria, with continuous effort by manufacturers to improve their performance quality [36]. However, none of these RDTs, are specifically targeted against *P. ovale* species, as most of the currently marketed RDTs target only *P. falciparum* and/or *P. vivax* or all human *Plasmodium* species, making it impossible to formally diagnose *P. ovale* spp. infection on this basis and thus resulting in misdiagnosis. Moreover, the sensitivity of existing RDTs is low (<50%) for *P. ovale* spp. because of low parasite density and/or structural variability of the targeted antigen [20,38,39,40]. To the best of our knowledge, no RDT till date has had the ability to specifically detect *P. ovale* spp., let alone differentiate between *Pow* and *Poc*.

Few studies have reported on RDT diagnostic efficiency in *Poc* and *Pow* but the number of cases were limited even in endemic regions. A recent study from France reported 677 *P. ovale* spp. imported cases (309 *Poc* and 368 *Pow*) and suggested a difference in sensitivity of the two species depending on the RDT type used for diagnosis [32]. Aldolase-RDT (47.8%) showed a higher sensitivity than pLDH-RDT (10.6%) for both species. In addition, two different RDT showed different sensitivity to detect *Pow*, i.e., 50% and 16% compared to 45.5% and 4.2% for *Poc* for Aldolase-RDT and pLDH-RDT, respectively; however, the difference was not significant for Aldolase-RDT [32]. The higher pLDH-RDT positivity with *Pow* may be explained by its ability to produce a higher amount of LDH compared to *Poc* [20]. Regardless of RDT type the sensitivity was density dependent for both species as previously reported [38,40] though pLDH-RDT would be better for *Pow* prevalence screening.

### 4.2. Microscopy

Microscopy remains the gold standard for malaria diagnosis and WHO recommends RDT and microscopic diagnosis for all suspected malaria cases [41]. Morphologically identical, *Poc* and *Pow* are indistinguishable even by expert microscopists and mostly misidentified as *P. vivax* [42].

Comparative studies of ultrastructural changes in erythrocytes infected with human *Plasmodium* species including *P. ovale* spp. have been carried out, providing some species distinguishing elements upon microscopic diagnosis, among which Schüffner’s stippling is present in *P. vivax* and *P. ovale* spp. but not in *P. falciparum* and *P. malariae* [43]. This approach is still used to distinguish *P. ovale* spp. from *P. malariae* and useful for *Poc* and *Pow* too. Another study including microscopic analysis of 48 samples (21 *Poc* and 27 *Pow*) revealed that all *Poc* and 19 *Pow* samples had Schüffner’s stippling and only eight *Pow* samples lacked them, indicating that Schüffner’s stippling only cannot be a strong discriminating feature between *Poc* and *Pow* [44].

### 4.3. Molecular Diagnostic Assays

Currently, only molecular techniques, in particular polymerase chain reaction (PCR) based assay, can specifically identify and distinguish *Poc* and *Pow*. Several genes depending on their level of sequence conservation are targeted to detect and distinguish between *Poc* and *Pow* infection. Commonly used genes in PCR assays are18S rRNA gene, tryptophan rich antigen gene, dihydrofolate reductase thymidylate synthase and sporozoite stage protein gene currently [14,20,33,45].

#### 4.3.1. 18S rRNA Gene

The 18S rRNA is the most common targeted gene for confirming the *P. ovale* spp. [46]. There are reports, which show polymorphism in 18S rRNA gene in *Poc* and *Pow* leading to PCR failure in few isolates [14]. Several other genes have also been used for subtyping *P. ovale* spp. which indicates *Poc* has greater genetic diversity as compared to *Pow* (Table 1) [33].

#### 4.3.2. Tryptophane Rich Antigen (tra) Gene

Some studies have targeted *tra* gene in PCR assay for specific discrimination between *Poc* and *Pow* [14,33,47] and also for the identification of genotypic variants within these two species [14]. Out of 22 *P. ovale* spp. samples, two failures were recorded in Kenya using *tra* gene compared to three with18S rRNA gene [47]. This indicates that *tra* gene can be a potential PCR target for the simultaneous identification and characterization of variants between ovale species.

#### 4.3.3. Dihydrofolate Reductase Thymidylate Synthase (*dhfr*) Gene

Using the *dhfr* gene as a target in PCR diagnosis, about 30 non-synonymous mutations as well as different amino acid repeats were identified in the *P. ovale* spp. *dhfr* gene for easy distinction of *Poc* from *Pow* than other human *Plasmodium* species [14,33,48].

#### 4.3.4. Sporozoite Stage Protein Gene

Sporozoite stage protein such as circumsporozoite protein/thrombospondin-related anonymous related protein (*cTrp*) and circumsporozoite surface protein (*csp*) genes, conserved genes, have also been reported by Saralamba and colleagues for *Pow* and *Poc* identification [45]. Further, these authors highlight distinct repeat regions in the *csp* protein in *Poc* (NPPAPQGEG) and *Pow* (DPPAPVPQG) as potential targets as monoclonal antibodies that can serologically distinguish sporozoites from these species in future studies.

#### 4.3.5. *Kelch 13* Gene

Recently, few authors have used the *Kelch 13* gene to distinguish the two ovale species but this gene seems to be less discriminant than *cTrp*, *csp* and merozoite surface protein 1 (*msp1*) genes [34].

#### 4.3.6. Other Method

A post quantitative polymerase chain reaction analysis method called high resolution melting (qPCR-HRM) was developed for *Poc* and *Pow* genetic identification based on their DNA melting temperature (Tm) [49]. Specific melting peaks have been identified at 71.26 °C and 71.04 °C respectively for *Poc* and *Pow* allowing their distinction.

Although PCR molecular identifications are more sensitive and specific than microscopy and RDT they have certain limitations like, the primers used in existing PCR assays are unable to specifically amplify all field isolates due to emerging genetic diversity [46]. In addition, in certain cases, the primers do not match with *P. ovale* spp. variants (*Pow*) resulting in cross reactions with other *Plasmodium* species, mostly with *P. vivax* [46]. Thus, the diversity of sequence and primers used could significantly influence the outcome of PCR reactions resulting in compromising the quality of diagnostic results and such epidemiological data regarding *Pow* and *Poc* can be misleading at times. 

**Table 1 diagnostics-11-01900-t001:** Some current genes used for molecular identification of *P. ovale* spp.

Subcellular Location	Genes	Accession Number	Putative Function	References
**Genome**	18S small subunit ribosomal RNA gene (ssrRNA)	PocGH01_00195420 *	RNA involves in proteins Translation	[8,14,20,33,46]
	Tryptophan rich antigen *(P*otra*)*	PocGH01_12010200 *	Protein involved in the parasite growth	[14,29,33]
	Reticulocyte-binding protein 2 *(P*orbp2*)*	PocGH01_00018800 *	Protein involved in the binding of the parasite to erythrocytes	[14,29,33]
	Circumsporozoite protein/thrombospondin-related anonymous related protein (cTrp)	PocGH01_08033300 *	Host cell surface receptor binding	[45]
	Circumsporozoite surface protein (csp)	PocGH01_00239700 *	Surface protein involved in the binding of the parasite to host hepatocytes	[45]
	Merozoite surface protein (msp1)	PocGH01_07037900 *	Surface protein involved in erythrocyte invasion	[45]
	Glyceraldehyde-3-phosphatase gene (Pog3p)	GU723542 ¶	Enzyme of the glycolysis pathway	[14,33]
	Dihydrofolate reductase thymidylate synthase gene (dhfr-ts)	PocGH01_05028400 *	Enzyme responsible for the production of folates and thymidylate	[14,33,44]
	Adenylosuccinate lyase gene (asl)	PocGH01_04023200 *	Enzyme of AMP biosynthesis pathway	[33]
	Kelch 13 gene	PocGH01_12019400 *	Protein involves in homooligomerization (substrate adaptors for cullin 3 ubiquitin ligases)	[34]
**Mitochondria**	Cytochrome c oxidase 1 gene (Cox 1)	PocGH01_MIT000030 *	Respiratory enzyme	[33]
	Cytochrome b gene (cytb)	PocGH01_MIT000020 *	Respiratory enzyme	[14,33]
**Apicoplast**	Factor Tu gene (tufA)	PocGH01_API002800 *	Translation elongation	[33]
	The caseinolytic protease C gene (clpC)	NA	Regulation of proteolytic complex	[33,49]

(*): gene ID from Plasmo DB data base, (¶): gene ID from GenBank data base, NA: Not available.

## 5. Biological and Clinical Comparison of *Poc* and *Pow*

In addition to their morphological identity, *Poc* and *Pow* share similar life cycle, biological and clinical characteristics [30]. *Poc* and *Pow* display same parasitaemia, symptoms, glycaemia, leucocyte count, haemoglobin, aspartate aminotransferase, alanine aminotransferase, bilirubin and albumin levels in infected individuals [20,29,31,32,44]. Though a multi-centric study conducted in Europe reported a significant low haemoglobin and creatinine level in *Poc* infected patients compare to *Pow* [20]. The potential differences in pathogenicity between *Poc* and *Pow* are not yet clearly established. However, some consistent observations seem to be relevant as they are reported by several independent studies (Table 2). Thrombocytopenia (platelet count < 150 × 10^3^/mm^3^) is a common complication in malaria infection, which can reach 94% [50]. A significantly deeper thrombocytopenia in patients with *Pow* malaria has been reported several times, as well as a shorter latency compared to those infected by *Poc* [20,29,31,32]. In addition, another study reported a higher temperature and a low albumin level in patients with *Pow* infection but the difference was not significant [31].

Rojo-Marcos and colleagues show a significantly higher frequency of *Pow* malaria infection in males (72.7%) compared to females (48.6%) especially in Caucasians group in a study including 35 *Poc* and 44 *Pow* cases [20]. In contrast, a recent report including eightfold larger sample size (309 *Poc* and 368 *Pow*) demonstrated no significant difference for gender, age or ethnicity [32] as earlier reported studies [29]. All these observations suggested that the variant type *Pow* could be more pathogenic as compared to the classic type *Poc*, obviously but this needs more future studies to be conducted on this aspect for efficient conclusive results.

## 6. Prophylaxis, Treatment and Relapse

Generally, prophylaxis by chemoprevention consists of taking an antimalarial drug (Chloroquine, Proguanil, Atovaquone/proguanil, Mefloquine, Doxycycline, Primaquine) recommended for anyone traveling to a malaria-endemic region [51]. New findings question the effectiveness of chemoprevention in case of *P. ovale* species. Many failures of malaria chemoprevention have been recorded in travellers returning from endemic countries [20,29,52,53]. Thirty six percent of chemoprevention failure was recorded in *P. ovale* spp. infections in British travellers compared to 6.4% and 23.7% for *P. falciparum* and *P. vivax* respectively [29]. Gallien et al. report a case of chemoprophylaxis failure using atovaquone-proguanil and another study of malaria surveillance from United States of America and Israel showed that 73% of *P. ovale* spp. malaria case occurred in travellers on effective prophylaxis treatment [52,53].

Another multicentre study conducted in Europe did not find any difference between those who used chemoprevention to those who did not; highlighting the ineffectiveness of chemoprevention in both *Poc* and *Pow* infection as none of molecules used can kill liver-stage hypnozoites [20]. Although the pathophysiological mechanism leading to dormant stage formation and relapse in *Poc* and *Pow* is not yet clearly established and is supported only by indirect arguments [54]. Nabarro and colleagues analysed the potential relationship between geographical and seasonal relapse in *P. ovale* spp. [30]. The authors highlighted the existence of a synchronized relapse in *Pow* with the malaria season in West Africa suggesting that hypnozoites would be programmed to activate at the time when the probability of being transmitted to a mosquito vector is highest [30]. This observed seasonal relationship in *Pow* does not exist for *Poc*.

In *P. ovale* spp. management therapy, drugs for both stages (blood and liver) should be given to the patients [55]. As in other *Plasmodium* species, the use of ACT is also recommended for *P. ovale* spp. malaria infection (Doxycycline, chloroquine, Artesunate, Artemether, Mefloquine, Quinine-Doxycycline, Artemether-Lumefantrine, atovaquone-Proguanil and Arteminol-Piperaquine) for successful blood parasite clearance [20,29,32].

## 7. Conclusions

*Poc* and *Pow* are two sympatric species that are almost identical biologically, physiologically and clinically. According to the existing data *Pow* seems to be the more pathogenic than *Poc* but this evidence remains to be scientifically established. The development of a simple, affordable and efficient tool for *Poc* and *Pow* differential diagnosis is a gap to be filled and should be seriously take into account in future investigations to eliminate this constraint in the fight against malaria. To date, molecular techniques remain the most accurate method to distinguish *Poc* and *Pow*.

## Figures and Tables

**Table 2 diagnostics-11-01900-t002:** Some significantly different reported parameters between *Poc* and *Pow*.

Parameter	Unite	Value (*n*)		Reference
		*Poc*	*Pow*	
**Thrombocytopenia**	Platelet count, median cells/μL	*126 (n = 21)*	*91.5 (n = 14)*	[31]
		*130 (n = 35)*	*105 (n = 44)*	[20]
		*111 (n = 309)*	*94 (n = 368)*	[32]
**Latency Period**	Geometric mean	85.7 *(n = 74)*	40.6 *(n = 60)*	[29]
	Median	86 *(n = 112)*	23 *(n = 131)*	[30]
	Median	72 *(n = 309)*	34 *(n = 368)*	[32]
**LDH Level**	Median value × IU/L	*267 (n = 35)*	*370 (n = 44)*	[20]
**Creatinine Level**	Median mg/dL	*0.79 (n = 35)*	*0.95 (n = 44)*	[20]
**Hemoglobin Level**	Median g/dL	*12.1 (n = 35)*	*13.2 (n = 44)*	[20]

*Poc: Plasmodium ovale curtisi*; *Pow: Plasmodium ovale wallikeri*; LDH: lactate dehydrogenase; *n*: number of samples included.

## Data Availability

Not applicable.

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
