# Peer review of "Epidemiological, Physiological and Diagnostic Comparison of Plasmodium ovale curtisi and Plasmodium ovale wallikeri"

_diagnostics, 2021, doi:10.3390/diagnostics11101900_

Round 1

Reviewer 1 Report

This mini-review article reports on the subject of the difference between Plasmodium ovale curtisi (Poc) and Plasmodium ovale wallikeri (Pow). The two species Poc and Pow are indistinguishable by microscopy but seem to differ in several points such as a duration of latency and pathogenicity. This review article presents and argues the several important information on Poc and Pow not only for advances in detection but also biological, epidemiological and clinical comparison. The article is presented in a reasonably clear manner. However, much improvement can still be made to make it clearer and more concise. Specific comments are as follows:

Major comments

  1. It is required to change the abstract due to less mention about the importance of distinguish Poc and Pow, it contains lots of back ground information about general malaria. Line 8-13 is a general malaria information, it is not required on abstract, and on the other hand the importance of diagnostic comparison appears to be only on last line17-18. On the current version of abstract is hardly to see the main information of this review article. It is strongly recommended the authors to revise the abstract on this MS.   
  1. It is better to clarify the current summary of molecular diagnostic assays for Poc and Pow. It is found each gene information and articles, but difficult to find the best (or better) option for PCR. The summary of PCR is required on the start part of 4.3 molecular diagnostic assays.
  1. Table 1 is quite useful information, however, it is required to add gene ID on each target genes. The audience are going to use these information for future experiments, nowadays the genome database on websites (such as a plasmodb) are quite nicely to organize with lots of upcoming information. If the author could show each gene ID (Poc and/or Pow) are going to be very useful. It is strongly recommended to add these information on table 1.

Minor comments

A) There are several strange fonts on this MS, line 116 (RDT and microscopy for all suspected malaria cases) and line 138 (tryptophan rich antigen gene). Please make it correctly.

B) Please confirm Ref 55 is correct for the references for PCR on each genes?? I guess it is not it.

Author Response

Major comments

  1. It is required to change the abstract due to less mention about the importance of distinguish Poc and Pow, it contains lots of back ground information about general malaria. Line 8-13 is a general malaria information, it is not required on abstract, and on the other hand the importance of diagnostic comparison appears to be only on last line17-18. On the current version of abstract is hardly to see the main information of this review article. It is strongly recommended the authors to revise the abstract on this MS.   

Reply: Abstract was reedited (line 33-42)

  1. It is better to clarify the current summary of molecular diagnostic assays for Poc and Pow. It is found each gene information and articles, but difficult to find the best (or better) option for PCR. The summary of PCR is required on the start part of 4.3 molecular diagnostic assays.

Reply: One paragraph was added (line 170-175). The best target gene for PCR detection and distinction of Poc and Pow is tra gene mentioned in section 4.3.2.

  1. Table 1 is quite useful information, however, it is required to add gene ID on each target genes. The audience are going to use these information for future experiments, nowadays the genome database on websites (such as a plasmodb) are quite nicely to organize with lots of upcoming information. If the author could show each gene ID (Poc and/or Pow) are going to be very useful. It is strongly recommended to add these information on table 1.

Reply: Done. Gene ID is added on each target genes in Table 1.

Minor comments

  1. A) There are several strange fonts on this MS, line 116 (RDT and microscopy for all suspected malaria cases) and line 138 (tryptophan rich antigen gene). Please make it correctly.

Reply: Done

  1. B) Please confirm Ref 55 is correct for the references for PCR on each genes?? I guess it is not it.

Reply: Reference 55 is not appropriate in table 1, it was removed.

Reviewer 2 Report

The manuscript presented by Hawadak et al. presents important aspects in the context of malaria caused by Plasmodium ovale, in addition to its specific variations. I think the manuscript is a brief review of the state of the art in the setting of the diagnosis of malaria caused by Plasmodium ovale. Due to the paucity of information in the literature, I think the manuscript could contribute to the development of the state of the art in the context of Plasmodium species causing human malaria.
I would like to make two minor suggestions/corrections:

1. Revise the italicized names of Plasmodium species presented in the manuscript (for example: Plasmodium vivax is not italicized in line 118);
2. I suggest that Table 1 be presented on a single page.

Author Response

Reply to Reviewer 2

Comments and Suggestions for Authors

The manuscript presented by Hawadak et al. presents important aspects in the context of malaria caused by Plasmodium ovale, in addition to its specific variations. I think the manuscript is a brief review of the state of the art in the setting of the diagnosis of malaria caused by Plasmodium ovale. Due to the paucity of information in the literature, I think the manuscript could contribute to the development of the state of the art in the context of Plasmodium species causing human malaria.
I would like to make two minor suggestions/corrections:

  1. Revise the italicized names of Plasmodium species presented in the manuscript (for example: Plasmodium vivax is not italicized in line 118);

Reply: Done. The entire manuscript was checked and the needful was done.

  1. I suggest that Table 1 be presented on a single page.

Reply: This remark will certainly be taken into account by the editor during the formatting.

Round 2

Reviewer 1 Report

My comments are addressed on the revised manuscript.